# Autophagy in Tissue Repair and Regeneration

**DOI:** 10.3390/cells14040282

**Published:** 2025-02-14

**Authors:** Daniel Moreno-Blas, Teresa Adell, Cristina González-Estévez

**Affiliations:** Department of Genetics, Microbiology and Statistics, School of Biology and Institute of Biomedicine (IBUB), University of Barcelona, Av. Diagonal, 643, 08028 Barcelona, Spain; dmoreno@ub.edu (D.M.-B.); tadellc@ub.edu (T.A.)

**Keywords:** autophagy, regeneration, tissue repair, injury, senescence, stem cell, planarian

## Abstract

Autophagy is a cellular recycling system that, through the sequestration and degradation of intracellular components regulates multiple cellular functions to maintain cellular homeostasis and survival. Dysregulation of autophagy is closely associated with the development of physiological alterations and human diseases, including the loss of regenerative capacity. Tissue regeneration is a highly complex process that relies on the coordinated interplay of several cellular processes, such as injury sensing, defense responses, cell proliferation, differentiation, migration, and cellular senescence. These processes act synergistically to repair or replace damaged tissues and restore their morphology and function. In this review, we examine the evidence supporting the involvement of the autophagy pathway in the different cellular mechanisms comprising the processes of regeneration and repair across different regenerative contexts. Additionally, we explore how modulating autophagy can enhance or accelerate regeneration and repair, highlighting autophagy as a promising therapeutic target in regenerative medicine for the development of autophagy-based treatments for human diseases.

## 1. Introduction

Autophagy is a highly conserved eukaryotic catabolic process responsible for the lysosomal degradation of intracellular components [1]. In mammals, three distinct pathways facilitate the delivery of cytosolic components to lysosomes for degradation. These include microautophagy [2], chaperone-mediated autophagy (CMA) [3], and macroautophagy (referred to here as autophagy) [4], the most extensively characterized form of autophagy (Figure 1). Autophagy can also be categorized based on the specific intracellular components targeted for degradation. These include xenophagy (clearance of intracellular pathogens), mitophagy (removal of mitochondria), ER-phagy (turnover of endoplasmic reticulum), aggrephagy (elimination of protein aggregates), lysophagy (degradation of lysosomes), lipophagy (breakdown of lipid droplets), and glycophagy (degradation of glycogen), among others. Each of these cargo types is recognized and sequestered by specific receptor proteins, which facilitate their selective targeting for autophagic degradation [5]. 

The process of autophagy initiates with the formation of an isolation membrane (also known as phagophore) in the cytosol. This membrane elongates to engulf portions of the cytoplasm, including proteins, damaged organelles such as mitochondria, the Golgi apparatus, endosomes, and even pathogens. Once the phagophore completely encloses the cargo, it forms an autophagosome, a double-membrane vesicle. The autophagosome then fuses with a lysosome to create an autolysosome, where the sequestered cytoplasmic material and the inner membrane of the autophagosome are degraded by lysosomal hydrolytic enzymes (Figure 1). Each stage of this intricate process is tightly regulated by a series of autophagy-related proteins (ATG proteins), which sequentially control phagophore formation, autophagosome biogenesis, and the fusion of the autophagosome with the lysosome [6].

Autophagy is an intracellular quality control and repair process that safeguards genomic stability and cellular integrity by removing damaged organelles, aggregated proteins, defective cytoplasmic components, and invading pathogens [7]. This degradation system participates in multiple cellular processes essential for the functional maintenance of normal tissues and organismal health, including cell death, preservation of stem cells, tumor suppression, and longevity [8]. Thus, as a key mechanism for cellular repair and renewal, defects in autophagy activity are closely linked to the decline of tissue homeostasis and a reduced regenerative capacity [9,10]. Despite growing interest in autophagy, comprehensive reviews specifically addressing its role in the complex mechanisms governing tissue regeneration remain limited. This review aims to address this gap by providing evidence supporting the involvement of autophagy in key biological processes underlying tissue repair and regeneration, including sensing the injury, immune response, peripheral barrier restoration, cell activation and cell cycle re-entry, differentiation and morphogenesis, and remodeling and scaling (Table 1). Particular emphasis is placed on findings from diverse regenerative species, offering a broader perspective on how autophagy could influence tissue regeneration across various biological contexts.

## 2. Molecular Machinery of Autophagy

To date, a number of autophagy-related proteins (ATGs) have been identified as controlling the complex molecular signaling of autophagosome biogenesis and have been described in detail elsewhere [6,24,25,26]. Here, we provide a brief overview of some of the most critical proteins involved in this process (Figure 2).

Phagophore formation is initiated by the activation of the ULK complex, comprising ULK1, ATG13, FIP200, and ATG101, in response to stress conditions, such as starvation, hypoxia, oxidative stress, protein aggregation, endoplasmic reticulum (ER) stress and others. This activation triggers a series of ULK1-dependent phosphorylation events, subsequently activating the class III phosphatidylinositol-3-kinase (PI3KC3) complex, which includes Beclin 1, Ambra1, and VPS34. PI3KC3 activity generates phosphatidylinositol 3-phosphate (PI3P), a phospholipid that facilitates the recruitment of DFCP1, ATG12, WIPI1, and additional ATG proteins, which together support phagophore elongation (Figure 2). At this stage, ATG5 is covalently conjugated to ATG12 through a ubiquitin-like reaction mediated by ATG7 and ATG10. The resulting ATG5-ATG12 conjugate then associates with Atg16L1, forming the Atg16L1 complex. Concurrently, the protease ATG4 cleaves LC3, producing LC3-I. LC3-I is subsequently conjugated to phosphatidylethanolamine (PE) through the coordinated actions of ATG7, ATG3, and the Atg16L1 complex, generating LC3-II. This form of LC3 is essential for autophagosome formation and is specifically localized to the autophagosome membrane.

Upon completion of autophagosome formation, ATG4 cleaves LC3-II from the outer autophagosome membrane, resulting in LC3-II localization specifically to the inner membrane of the autophagosome. Subsequently, autophagosomes are transported along microtubules, with SNARE proteins facilitating their fusion with lysosomes. Finally, the lysosomal hydrolases degrade the cytoplasmic material enclosed within the autophagosome, and the resulting biomolecules are released back into the cytosol to be used again by the cell (Figure 2).

The autophagy machinery can be initiated by a wide range of endogenous and exogenous stimuli, including nutrient deficiency, inflammation, hypoxia, reduced cellular energy levels (e.g., ATP depletion), and pathogenic infections [27]. This dynamic and tightly regulated process is also activated during tissue regeneration and wound healing, functioning as an indispensable mechanism promoting tissue repair and restoration.

## 3. Regeneration and Repair

The regeneration process refers to the precise replacement of tissues, organs, or body parts following injury, aimed at restoring the original structure and function [11]. The ability to regenerate varies across the animal kingdom, ranging from species that can fully reconstruct entire organisms from small fragments (whole-body regeneration), for example, planarians and cnidarians, those that can regenerate full organs as seen in fish and salamanders, to the ones that have a more restricted capacity of regeneration limited to certain tissues or cell types, for example, humans (Figure 3) [28]. In contrast, repair can restore some aspects of the original tissue structure, but it may also lead to structural abnormalities, such as scar formation or fibrosis, which can negatively affect organ function. Thus, regeneration and repair constitute complex phenomena that present challenges in cross-species and cross-tissues comparison.

The most important biological processes that facilitate regeneration and also aid in repair include: (1) Sensing the injury, (2) Immune response, (3) Peripheral barrier restoration, (4) Cell activation (may involve migration), cell cycle re-entry and proliferation, (5) Differentiation and morphogenesis, and (6) Remodeling and scaling (Table 1) [11,29,30,31,32,33,34]. While these processes operate in concert to achieve full tissue repair and restoration, their manifestation can vary significantly between species/tissues with high regenerative capacities and those with more limited regenerative potential or those that can only repair damage.

In this review, we focus on the key biological processes involved in tissue regeneration and repair, considering various types of damage, including microbial infections, treatments with ligands and agonists such as LPS, exposure to toxins, skin injuries, limb amputations, and neuronal degeneration, all of which necessitate repair or regeneration. We will explore how autophagy influences various of these biological processes (Figure 4) and how this modulation can impact regeneration and repair. Additionally, we will discuss its potential as a therapeutic target in regenerative medicine.

## 4. Exploring a Possible Role of Autophagy in Sensing and Responding to Injury

Following injury, wounded tissues initiate a coordinated response aimed at promoting repair, preventing infection, clearing cellular debris, restoring cell populations, and reorganizing tissue architecture [35]. This response is driven by rapid molecular changes in cells at the wound site, which activate critical signaling pathways for wound detection and initiating early responses. Key signals released during this phase include damage-associated molecular patterns (DAMPs) such as extracellular adenosine triphosphate (ATP), reactive oxygen species (ROS), polyunsaturated fatty acids (PUFAs), and increase in intracellular Ca^2+^ [35,36]. Each of these damage signals can initiate the autophagic pathway or be modulated by the autophagy machinery. In this section, we will specifically examine the interplay between DAMPs, ATP, and ROS in regulating autophagy within the context of tissue regeneration.

### 4.1. Autophagy and DAMPs in Tissue Regeneration

***Release of DAMPs by Autophagy***. Upon tissue injury and predominantly observed in vertebrates, a group of cellular molecules, including nuclear and cytosolic proteins, known as damage-associated molecular patterns (DAMPs) are released by stressed, injured, or dying cells. These molecules are subsequently recognized by specialized receptors on immune cells, triggering critical signaling pathways essential for initiating tissue repair. Relevant DAMPs include the nuclear protein high mobility group box 1 (HMGB1), an abundant chromatin-binding protein and one of the best-characterized DAMPs [37,38,39]. Another example is adenosine triphosphate (ATP), the primary bioenergetic substrate and metabolic currency, which can also be released into the extracellular space where it functions as a signaling molecule regulating many biological processes [37,40,41,42]. Research has shown that autophagy regulates the selective release and secretion of HMGB1 and ATP in cells destined to die. For instance, studies in vitro using human tumor cells treated with diphtheria toxin, which kills tumor cells, showed that dying cells with elevated autophagy selectively release HMGB1 without disrupting the cell membrane, whereas cells with blocked autophagy retain HMGB1 [43]. A recent study using human and murine cell lines revealed that autophagosome formation is crucial for HMGB1 secretion under stress conditions such as starvation, and phorbol 12-myristate 13-acetate (PMA) and trichostatin A (TSA) treatments, with HMGB1 secretion notably inhibited by an early autophagy inhibitor and reduced in ATG5-deficient cells, highlighting the involvement of the secretory autophagy machinery in HMGB1 release [44]. In addition, using the Ba/F3 mouse cell line, it was observed that dying (induced by doxorubicin) autophagic cells release ATP that stimulates inflammasome activation in macrophages and secretion of interleukin-1 beta (IL-1β) [45]. Consistent with this, research indicates that autophagy is essential for the release of ATP from dying mouse tumor cells, which stimulates antitumor immune responses [46]. An additional study using HeLa cells demonstrated that under starvation conditions ATP is released to the extracellular space via autophagic vesicles in a vesicle-associated membrane protein 7 (VAMP7)-dependent manner [47]. Collectively, these investigations emphasize the role of autophagy in mediating DAMPs release under various conditions, which may play a crucial role in sensing and responding to tissue injury. However, no single study has directly demonstrated the contribution of autophagy-dependent DAMP release in the contexts of tissue repair or regeneration. Exploring this connection further could provide valuable insights for future research in the regenerative field.

Autophagy activation by DAMPs. The relationship between DAMPs and autophagy could be bidirectional. HMGB1 and ATP may also activate the autophagic process. Research in cultured mouse embryonic fibroblasts (MEFs) has shown that cytosolic HMGB1 disrupts the Beclin1–Bcl-2 interaction by competitively binding to Beclin1, leading to the dissociation of the Beclin1–Bcl-2 complex and the subsequent induction of autophagy under starvation [48]. Additionally, HMGB1 activates autophagy in response to oxidative stress, as evidenced by findings that inhibition of HMGB1 release or loss of HMGB1 in MEFs results in reduced LC3-II formation, a decrease in autophagosome formation, and blockage of autophagic flux under oxidative stress conditions [49]. Moreover, a recent study in vitro showed that HMGB1 induces autophagy in mouse macrophages in response to lipopolysaccharide (LPS) [50]. Another line of research provided evidence that ATP induces rapid cell autophagy in human macrophages in vitro as an efficient mechanism to control mycobacterial infections [51]. Also, in cultured mouse microglial cells, it was observed that ATP treatment triggers the release of autolysosomes into extracellular spaces probably as a mechanism to eliminate the undigested material contained in lysosome-related organelles that can be harmful if they remain in the cytosol [52]. These studies suggest a mutual regulation between DAMPs and autophagy in different conditions. Despite this, the activation of autophagy by DAMPs has not been reported in the context of tissue repair or regeneration, which would be highly relevant for the field.

### 4.2. ROS and Autophagy: A Possible Interaction in Tissue Regeneration

Reactive oxygen species (ROS) are highly reactive metabolites derived from molecular oxygen (O_2_) and include nonradical hydrogen peroxide (H_2_O_2_), hydroxyl radical (·OH), and superoxide radical anion (O_2_−) [53]. At physiological levels, ROS function as signaling molecules that contribute to the regulation of various cellular processes [54]. Multiple lines of research support the idea that ROS plays a regulating role in early wound responses and regeneration across various animal models. For instance, studies indicate that amputation triggers a sustained ROS production as an early event essential for blastema formation and the progression of regeneration, as observed in tadpole tail regeneration in *Xenopus* [55] and during fin regeneration in adult zebrafish [56]. Similarly, ROS have been shown to activate key signaling pathways necessary for imaginal disc regeneration in *Drosophila* [57]. In planarians, amputation-induced ROS production is required for differentiation and successful regeneration [58,59]. Also, ROS signaling is considered one of the hallmarks of regulating human wound healing by affecting inflammation, cell proliferation, angiogenesis, and extracellular matrix formation [60]. 

Almost all autophagy steps from autophagosome induction and formation till autophagy maturation and cargo degradation are affected by ROS [61]. For example, elevated ROS levels result in the oxidation of Atg4, which triggers autophagosome formation [62]. ROS can regulate autophagy by activating AMPK, which subsequently phosphorylates the ULK1 complex, leading to the induction of autophagy [63,64]. Moreover, oxidation of TFEB (transcription factor EB), a master transcription factor of autophagy, enables TFEB to translocate to nuclei for active transcription of autophagic and lysosomal genes to control autophagy [65]. Collectively, these evidences suggest that injury-induced ROS may play a critical role in activating the autophagy pathway, which can be crucial for the next steps of the regeneration process.

On the other hand, although autophagy is modulated by ROS, autophagy also exerts a feedback mechanism to control ROS levels. This occurs through the degradation of ROS-generating organelles like mitochondria (mitophagy) [66] and proteins that are involved in ROS generation [67]. In this context, studies have shown that autophagy plays a key role in tissue regeneration by modulating ROS levels. For instance, autophagy has been found to support irradiation-induced intestinal regeneration in *Drosophila* by reducing excessive ROS in intestinal stem cells [68]. Similarly, mitophagy activation is essential for functional regeneration of skeletal muscle following myotoxic injury in mice by cardiotoxin injection into the left tibialis anterior muscles [69]. However, inducing mitophagy to reduce mitochondrial ROS levels has been shown to impair liver repair after partial hepatectomy [70]. Together, these findings highlight the complex interplay between ROS and autophagy, suggesting that regulation of either component can significantly influence tissue regeneration outcomes.

## 5. Functional Interplay Between Autophagy and Immune Defense Responses

Tissue injury in vertebrates generally triggers an immune response, which plays a crucial role in tissue repair and regeneration [71]. Upon tissue injury, an inflammatory response is triggered by DAMPs released from dying cells and pathogen-associated molecular patterns (PAMPs) from invading organisms. This response provides an immediate defense against potential pathogens infiltrating the injured tissue. Toll-like receptors (TLRs) and other pattern recognition receptors (PRRs) recognize these danger signals and initiate a complex inflammatory cascade. This cascade activates tissue-resident macrophages, which subsequently facilitate the recruitment of neutrophils, monocytes, macrophages, and other immune cells. Together, these cellular components orchestrate a coordinated immune response critical for effective tissue repair. For a comprehensive overview of the key players involved in the immune response following tissue injury and during regeneration, detailed discussions are available in previously published reviews [72,73,74,75].

In response to damage signals, several pattern recognition receptors (PRRs) and inflammatory proteins can activate autophagy to support antimicrobial defense and modulate inflammatory responses. For instance, in murine and human macrophage cell lines treated with lipopolysaccharide (LPS), Toll-like receptor 2 (TLR2) and TLR4 induce autophagy by stabilizing Beclin 1, which is crucial to initiate autophagosome formation [76,77]. NOD1 and NOD2, two well-characterized pattern recognition receptors (PRRs) of the NOD-like receptor (NLR) family, activate autophagy in human and murine cell lines after bacterial sensing by recruiting ATG16L1 to the plasma membrane at bacterial entry sites. This process facilitates the encapsulation of invading bacteria by autophagosomes, leading to their subsequent degradation [78,79]. In addition, stimulation of TLR2/6 or TLR4 with TLR ligands induces autophagy in primary human keratinocytes to prevent excessive inflammation and modulate the inflammatory response [80]. Another study demonstrated that Toll-like receptor 2 (TLR2) promotes autophagy in renal tubular epithelial cells via the PI3K/Akt signaling pathway following cisplatin-induced acute kidney injury [81]. In this context, skin wounds in mice induce autophagy in keratinocytes through TNF and NFκB activation. This autophagy, in turn, promotes wound repair by facilitating the recruitment of macrophages, neutrophils, and mast cells via the regulation of CCL2 (C-C motif chemokine ligand 2) expression [82]. Interestingly, studies using a UV-mediated acute skin injury mouse model and human skin biopsies revealed that vitamin D-induced autophagy in macrophages improves wound healing by facilitating macrophage differentiation [83]. Conversely, immune machinery-induced autophagy can serve as a feedback mechanism to suppress inflammation by degrading pattern recognition receptor (PRR) pathway sensors and other key inflammatory proteins. For instance, using human and mouse cells, it was found that the TRIF adaptor's selective autophagic degradation is required to terminate the TLR3/4-mediated innate immune and inflammatory responses after stimulation with various TLR ligands [84,85]. Inflammasomes are cytosolic signaling complexes that drive inflammation in response to damage signals detected by PRRs. Research involving mouse models, as well as human and mouse cell lines, indicates that autophagic degradation of inflammasomes helps regulate inflammation upon stimulation of macrophages with NLRP3 agonists [86,87]. In addition, it has been shown that autophagy controls the secretion of the proinflammatory cytokine interleukin-1β (IL-1β) by targeting pro-IL-1β for lysosomal degradation and by regulating the activation of the NLRP3 Inflammasome in mice after treatment with LPS [88]. Taken together, these studies highlight the intricate interplay between autophagy and the immune response in vertebrates, underscoring the important role of autophagy in the clearance of intracellular pathogens and the regulation of immune defense mechanisms. Nevertheless, despite all the evidence presented, information on the role of autophagy in regulating these inflammatory factors during tissue regeneration or repair is extremely scarce. This presents an exciting avenue for future research. 

### The Crosstalk Between Autophagy and Inflammation During Skin Wound Healing

Inflammation is an important response that plays a crucial role in the regeneration of injured tissues [89,90]. As previously mentioned, tissue injury triggers the release of danger signals (DAMPs and PAMPs), which activate TLRs and PRRs, leading to inflammation through the activation of transcription factors such as NF-κB and interferon regulatory factors [75]. This regenerative inflammatory response is a crucial step in skin wound healing and begins with the activation of monocytes into pro-inflammatory M1 macrophages. Subsequently, these M1 macrophages undergo polarization into the anti-inflammatory and pro-regenerative M2 phenotype, a key process essential for proper tissue repair and healing [91]. In this context, evidence has shown that macrophage autophagy plays an essential role in macrophage polarization [92]. For example, a recent study demonstrated that the chemokine C-C motif ligand 6 (CCL6) promotes wound healing by increasing M2-type macrophage levels. This effect is mediated through the inhibition of macrophage autophagy via activation of the PI3K/Akt signaling pathway during skin wound healing in mice [93]. During the inflammatory phase of wound healing, a large number of cytokines and chemokines are secreted resulting in the recruitment of neutrophils that also produce cytokines such as tumor necrosis factor-a (TNF-α), IL-1β and IL-6 to amplify the inflammatory response [94,95]. In this context, a recent study demonstrated that both wounding and TNF induce the autophagic flux in keratinocytes through NF-κB activation. This autophagy activation is essential for the recruitment of macrophages, neutrophils, and mast cells to the wound site via the induction of CCL2 [82], a key chemokine involved in the migration and infiltration of monocytes and macrophages [96]. Interestingly, another study demonstrated that pharmacological inhibition of autophagy using 3-MA exerts anti-inflammatory effects by downregulating the expression of pro-inflammatory cytokines, including TNF-α, IL-1β, and IL-6, and promoting skin wound healing in mice [97]. Moreover, autophagy can directly influence the secretion of inflammatory molecules. It has been reported that autophagy stimulation in macrophages facilitates the release of proinflammatory factors, particularly IL-1β and HMGB1, through a process known as secretory autophagy [98,99]. This mechanism may contribute to reparative inflammation during wound healing. 

Overall, it is clear that autophagy significantly influences the early stages of the wound healing process. It is essential for initiating inflammation through multiple pathways and contributes to the mobilization of neutrophils, monocytes, and macrophages. Therefore, a deeper understanding of the interplay between autophagy and inflammation during wound healing could open new avenues for developing therapeutic strategies to enhance tissue repair and improve clinical outcomes.

## 6. Autophagy and Its Role in Regulating Cell Proliferation During Regeneration: Stem Cell Proliferation or Cell Cycle Re-Entry

A marked increase in mitotic activity is a characteristic feature observed across organisms with high regenerative capacity [16]. Following injury, regenerative processes can either enhance the cell cycle activity in pre-existing populations of dividing stem cells or induce differentiated cells to re-enter the cell cycle. In both scenarios, cell proliferation emerges as a crucial determinant of successful regeneration [16]. For instance, in the cnidarian *Nematostella*, stem cell proliferation significantly increases between 24- and 48-h post-amputation (hpa) during head regeneration [100]. In the acoel model *Hofstenia*, stem cell proliferation increases during regeneration as early as 6 hpa [101]. In the freshwater planarian, *Schmidtea mediterranea*, one of the most extensively studied model systems for whole-body regeneration, amputation triggers two peaks of increased stem cell mitosis in response to injury. The first mitotic peak, by 6 hpa, represents a body-wide response to any injury, while a second, by 48-72 hpa, is triggered only when the injury results in tissue loss [102]. Additional studies have shown an increase in cell proliferation in *Drosophila* imaginal wing disc regeneration [103], during zebrafish fin regeneration [104], the Mexican axolotl limb regeneration [105] and during skin wound healing [82].

Autophagy plays a complex role in controlling cell proliferation by modulating the availability and degradation of key cell cycle regulators. In mammals, through selective autophagic degradation, it influences cyclin-dependent kinases (CDKs), cyclin-dependent kinase inhibitors (e.g., CDKN1B/p27), and E2F transcription factors—essential players in cell cycle progression. Notably, this regulatory relationship is bidirectional: while autophagy modulates the activity and turnover of these cell cycle components, several of these factors, in turn, exert control over autophagic processes, creating a dynamic feedback loop [106,107]. This intricate interplay highlights the dual role of autophagy in both promoting and restraining cell proliferation, depending on the cellular context and stress conditions.

Emerging evidence highlights autophagy as a crucial process for maintaining the function and maintenance of various stem cell populations [108]. Autophagy supports stem cell survival and function by facilitating cellular remodeling, mitigating reactive oxygen species (ROS) production, and preventing DNA damage through the removal of damaged mitochondria [108]. For instance, impaired autophagy in muscle stem cells (satellite cells) results in increased mitochondrial dysfunction and ROS production, driving the cells into senescence and leading to defective muscle regeneration following myotoxic injury in mice by cardiotoxin injection [109]. Moreover, autophagy is critical for the activation of muscle stem cells, transitioning them from a quiescent state into the cell cycle. The autophagy process ensures the availability of essential nutrients and metabolic precursors necessary for stem cell activation and proliferation during mice skeletal muscle regeneration after injection of BaCl_2_ in the tibialis anterior (TA) muscle [110]. 

Numerous studies have highlighted the critical role of autophagy in regulating cell proliferation during regeneration across various biological systems. For example, autophagy has been shown to promote cell proliferation during the early stages of zebrafish fin regeneration [111]. In *Drosophila*, autophagy is essential in intestinal stem cells to sustain proliferation, enabling continuous regeneration of the intestinal epithelium after damage with dextran sodium sulfate, DSS [112]. Similarly, epidermal autophagy has been demonstrated to support keratinocyte proliferation, facilitating wound healing and skin repair [82]. Furthermore, the activation of autophagy with rapamycin in rats accelerated skin regeneration by enhancing cell proliferation and increasing the population of epidermal basal stem cells and hair follicle stem cells [113]. 

These studies provide insights into the important role of autophagy in supporting cell proliferation during tissue regeneration. 

A central mechanism accompanying tissue repair and regeneration observed in many organs and species is the process of cell cycle re-entry. Organisms with high regenerative capacity employ diverse and unique strategies to replace lost or damaged tissue, such as inducing terminally differentiated cells to revert to a less specialized state within their lineage (dedifferentiation), prompting cells to switch lineages to generate a different cell type (transdifferentiation) or simply inducing cell proliferation [114]. Notably, cell cycle re-entry is the common component of all these mechanisms involved in tissue regeneration. For example, tissue injury can stimulate cell cycle entry of cardiomyocytes during heart regeneration in zebrafish [115]. Similarly, in axolotls, connective tissue cells near the injury site dedifferentiate to form a blastema, which proliferates and redifferentiates to regenerate all components of the lost limb [116,117]. In mammals, Schwann cells dedifferentiate and proliferate following nerve injury to support nerve regeneration [118]. Interestingly, recent studies in mammalian stomach and pancreas have found that fully differentiated cells return to proliferation in an autophagy- and mTORC1-dependent manner [119,120]. In a model of high-dose tamoxifen injury repair in the stomach and pancreas, downregulation of mTORC1 promotes autophagy to degrade differentiated cell components and damaged organelles. Then, cells induce expression of wound-healing associated genes including *Sox9*, *Clu*, and *Cd44v*, while reactivating cellular metabolism. Subsequently, mTORC1 activity is restored, enabling cells to exit their differentiated state and re-enter the cell cycle to support tissue regeneration [119,120]. This regenerative mechanism is termed *paligenosis*, a term originating from the Greek words *palin* (indicating “recurrence” or “going backward”), *genea* (meaning “origin” or “producing”), and *osis* (denoting “a process or action”). It is regulated by two key genes, *Ddit4* and *Ifrd1* [121,122]. Similarly, during zebrafish muscle regeneration, autophagy activation early in the regenerative response to muscle injury facilitates myocyte dedifferentiation by regulating cytoplasmic remodeling after large myectomy [123]. Curiously, autophagy has also been shown to play a critical role in reprogramming differentiated cells into induced pluripotent stem cells (iPSCs). During the early stages of iPSC generation, the transcription factor Sox2 suppresses mTOR activity, leading to a transient increase in autophagy. This autophagic phase is essential for cellular reprogramming, after which mTOR activity is restored in later stages to ensure the successful completion of the reprogramming process [124]. Furthermore, nerve injury has been shown to activate the selective autophagic degradation of myelin (myelinophagy) in Schwann cells, facilitating their dedifferentiation which is important for nerve repair and regeneration [125,126]. In addition, autophagy has been reported to facilitate the dedifferentiation and recovery of the regenerative capacity of aged human bone marrow mesenchymal stem cells cultured in a three-dimensional system [127]. Collectively, these studies identify autophagy activation as a potential mechanism that could facilitate cell cycle re-entry, thereby promoting tissue repair and regeneration.

## 7. Autophagy Promotes Cellular Migration for Tissue Regeneration and Wound Repair

Following tissue injury, the recruitment and migration of different cell types to the wound site is essential for promoting tissue repair and regeneration. For example, adult stem cells (neoblasts) migrate to wound sites during planarian regeneration in response to tissue loss [102,128]. During zebrafish regeneration, neutrophils and macrophages have been observed to migrate to the wound site in response to both heart [129] and fin injuries [130]. Intestinal stem cells also migrate towards the wound within the *Drosophila* intestinal epithelium to facilitate the regeneration of the adult intestine after enteropathogenic infection and tissue damage by laser ablation [131]. In addition, during mammalian skin repair, epithelial cells migrate to contribute to wound closure and tissue repair [132]. In this line, accumulating evidence suggests that autophagy plays an important role in cell migration by regulating degradation of essential components involved in this process [133,134]. For example, autophagy has been shown to promote cell migration through the NBR1-dependent selective degradation of focal adhesion proteins in motile cells in a scratch-wound healing migration assay [135]. Additionally, autophagy induced by TLR signaling enhances cancer cell migration by stimulating the production of cytokines and chemokines necessary for their increased motility [136]. In breast cancer cells, the activation of autophagy has also been linked to enhanced migratory capacity [137,138]. Furthermore, autophagy regulators such as DRAM1 and p62 have been shown to play critical roles in controlling the migration and invasion of cancer stem cells [139]. In the context of tissue regeneration, the induction of autophagy is essential for keratinocyte migration, thereby facilitating wound healing in mice following injury [82,140]. Additionally, autophagy has been found to play an important role in the recruitment of blood cells to wound sites in *Drosophila* larvae and for the spreading of mouse macrophages after puncture wounding [141]. Recent studies have demonstrated that autophagy enhances nerve regeneration by promoting the migration of Schwann cells after nerve injury [142,143]. Furthermore, the activation of autophagy has been shown to stimulate the migration of dental pulp stem cells, contributing to pulp regeneration [144]. These studies suggest an important role of autophagy in regulating cell migration during wound healing and tissue repair. However, further research is needed to clarify the direct relationship between autophagy and cell migration in supporting regeneration across other organs in mammals and various experimental model organisms.

## 8. Autophagy in Tissue Remodeling and Scaling During Regeneration

The final phase of the regeneration process involves the perfect restoration of the shape, size, and proportions of the regenerating tissue, ensuring that it aligns seamlessly with the structure and function of the original tissue. Particularly in whole-body regeneration, this process involves the remodeling of pre-existing tissue and its integration with newly formed cells, culminating in the complete reestablishment of tissue morphology and functionality [20,145]. This relies on the coordinated interplay of various cellular mechanisms, such as cell death to shape the correct tissue structure [23,146,147,148,149], extracellular matrix (ECM) remodeling to support new tissue architecture [150,151,152], and the activation and regulation of the Hippo/YAP and Wnt pathways that guide cells in determining their positional identity and functional roles within the regenerating tissue [153,154,155,156]. Any dysregulation of these tightly orchestrated processes can result in incomplete or aberrant regeneration or tumorigenesis [156]. The autophagy pathway is widely recognized for its influence on all those processes. In studies in mammals, autophagy plays a crucial role in regulating cell death [157], contributes to the remodeling and deposition of extracellular matrix [158,159], and modulates key signaling pathways such as Hippo and Wnt signaling [160,161]. The critical role of autophagy in tissue remodeling in the context of regeneration has been well-documented in studies using planarians [162]. Studies in the planarian *Girardia tigrina* have highlighted the role of autophagy in cell death, particularly during planarian remodeling. The gene *Gtdap-1*—the ortholog of human death-associated protein-1 (DAP-1)—is specifically upregulated in regions and timeframes where remodeling occurs during regeneration, facilitating the proper scaling of the body. Additionally, *Gtdap-1* expression increases during starvation in the ovaries, testes, and copulatory apparatus of sexual planarians, organs known to regress under starvation conditions. Moreover, transmission electron microscopy revealed that *Gtdap-1* is expressed in cells exhibiting autophagic morphology. A subset of *Gtdap-1*-positive cells was also positive for cleaved caspase-3, indicating a link between autophagy and cell death. Furthermore, RNAi-mediated down-regulation of *Gtdap-1* reduced caspase-3 activity and cell proliferation levels, leading to remodeling defects in planarians [163,164]. Similarly, a recent study demonstrated that RNAi-mediated knockdown of *atg1* in planarians produces a regeneration phenotype comparable to that observed with *Gtdap-1* knockdown. This includes a reduction in both cell death and cell proliferation [165]. In line with this, in regenerating Hydra, silencing of the serine protease inhibitor *Kazal1* triggered excessive autophagy, resulting in severe tissue disorganization, massive gland cell death, and vacuolization of digestive cells. This suggests that autophagy dysregulation, as induced by *Kazal1* knockdown, disrupts tissue organization, impairs cellular homeostasis, and compromises the survival of cells subjected to amputation-induced stress [166]. Thus, by orchestrating cell death and modulating key signaling pathways involved in tissue remodeling, the precise regulation of the autophagy pathway is essential during the late phases of tissue regeneration. Dysregulation of autophagy, such as excessive autophagic activity, can disrupt tissue organization and compromise structural integrity, ultimately hindering the resolution of the regenerative process. Future studies using diverse animal and tissue models will elucidate the direct role of autophagy in maintaining proper tissue structure and organization during the resolution phase of regeneration. 

## 9. Connecting Autophagy and Senescence During Regeneration

Cellular senescence is a stress response characterized by a stable arrest of the cell cycle, accompanied by nuclear and cytoplasmic damage, as well as the secretion of a complex array of signaling molecules collectively known as the senescence-associated secretory phenotype (SASP) [167]. Senescence has traditionally been linked to aging and age-related diseases; however, accumulating evidence suggests that senescent cells also play essential physiological roles, including tumor suppression and contributions to embryonic development [168]. Furthermore, cellular senescence has recently emerged as a central component of the regeneration process in various regenerative species. For example, head amputation in cnidarians such as *Hydractinia* triggers senescence in a subset of cells near the injury site. These senescent cells secrete signals that promote the dedifferentiation and proliferation of neighboring cells, facilitating whole-body regeneration in these organisms [169]. In zebrafish, fin amputation causes the accumulation of senescent cells at the injury site, and pharmacological elimination of these cells has been shown to impair fin regeneration [170]. Similarly, during salamander limb regeneration, senescent cells accumulate at the wound site, where they secrete signals that promote cell cycle re-entry of neighboring cells, facilitating proper limb regeneration [171,172,173]. Additionally, senescent cells have been observed at wound sites following skin injury, where they contribute to promoting wound healing [174,175]. This evidence highlights cellular senescence as a key emerging factor in tissue regeneration and repair. Notably, autophagy has been shown to exhibit dual roles in the regulation of cellular senescence, functioning as both a pro-senescence and anti-senescence mechanism depending on its spatiotemporal dynamics [176,177]. For example, autophagy can prevent the induction of senescence by eliminating many of the stressors capable of inducing the senescence program, such as dysfunctional mitochondria and ROS [109]. Furthermore, selective autophagic degradation of specific proteins, such as the transcription factor GATA4, has been reported to inhibit senescence and suppress the expression of the SASP [178]. Conversely, autophagy has been shown to promote senescence by the selective degradation of nuclear components, including nuclear envelope proteins [179,180], and cytoplasmic chromatin fragments [181]. Additionally, autophagic activity has been observed to enhance the synthesis and secretion of SASP components during oncogene-induced senescence [182]. However, limited evidence exists regarding the interplay between autophagy and senescence in the context of tissue regeneration and repair. For instance, autophagy has been shown to prevent senescence in aged muscle stem cells by degrading dysfunctional mitochondria and mitigating oxidative stress [109]. Similarly, autophagy plays a key role in liver repair by preventing hepatocyte senescence after partial hepatectomy [183]. Senescent cells are increasingly recognized as key contributors to fibrosis, a pathological process characterized by excessive deposition of extracellular matrix (ECM) components due to dysregulation of normal wound-healing mechanisms [184]. Through the secretion of pro-fibrotic factors within the senescence-associated secretory phenotype (SASP)—notably transforming growth factor-beta (TGF-β), interleukin-11 (IL-11), and plasminogen activator inhibitor-1 (SERPINE1)—senescent cells actively drive fibrotic responses and contribute to the progression of fibrotic diseases [185,186,187,188,189]. Autophagy has also been implicated in the development of fibrosis [190,191,192,193]. However, research exploring the interplay between autophagy and senescence in fibrotic conditions remains limited. One study demonstrated that sustained autophagy induction is associated with enhanced mTORC2 activity and fibroblast senescence, which restricts myofibroblast differentiation and, consequently, may help prevent fibrosis [194]. Additionally, autophagy is known to influence the senescence-associated secretory phenotype (SASP) by promoting protein secretion through the formation of a specialized cellular compartment known as the target of rapamycin (TOR)-autophagy spatial coupling compartment (TASCC) [182]. In this context, autophagy has been shown to enhance the expression and secretion of fibrotic factors, likely through the SASP, in senescent tubular cells following acute kidney injury [195]. Consistent with this, another study demonstrated that autophagy promotes the secretion of profibrotic factors via the formation of the TASCC in senescent renal tubular cells after kidney injury, ultimately contributing to kidney fibrosis [196]. This highlights the intricate interplay between autophagy and senescence in tissue repair. A deeper understanding of their complementary roles could offer valuable insights into how the autophagy pathway promotes a pro-healing senescent state, facilitating organ repair and regeneration.

## 10. Autophagy and Lipid Metabolism During Tissue Regeneration

Lipid metabolism is crucial for tissue regeneration and repair, particularly in the liver [197] and muscle [198,199] and for cell differentiation during planarian regeneration [200,201]. Autophagy, in turn, plays an essential role in regulating lipid metabolism [202] and is particularly important for maintaining energy homeostasis, especially in the liver [183]. For instance, lipophagy facilitates energy production by breaking down stored lipids into free fatty acids, which are subsequently oxidized in mitochondria to generate ATP. This energy is critical for liver repair, as the process requires substantial energy to support cell proliferation and tissue growth after partial hepatectomy [183,203]. Studies have demonstrated that the inhibition of autophagy via *Atg*7 knockdown in hepatocytes leads to reduced lipid droplet accumulation and impairs liver repair following partial hepatectomy in mice [204]. Interestingly, the stimulation of autophagy in macrophages using trehalose reduced intracellular lipid droplet accumulation and enhanced repair and remyelination in models of CNS demyelination [205]. Additionally, studies in muscle have revealed that elevated lipophagy in *Prmt5* knockout mice leads to depletion of lipid droplets, impairing muscle regeneration following cardiotoxin (CTX)-induced injury [206]. While these findings suggest a critical role for lipophagy in tissue regeneration, direct evidence establishing a clear link between lipophagy and regenerative processes remains limited. Maintaining a delicate balance between lipid degradation and lipid droplet accumulation appears to be crucial for achieving optimal regeneration outcomes. Undoubtedly, the specific contribution of lipophagy to tissue regeneration represents an intriguing and promising area for future research, with the potential to deepen our understanding of how lipid metabolism and autophagy coordinate to drive effective tissue repair and regeneration.

## 11. Autophagy as a Therapeutic Target for Tissue Repair and Regeneration

Considering the important role of autophagy in regulating various cellular functions, and its significant involvement in human health and diseases, a diverse range of activators and inhibitors has been developed to modulate the autophagy activity under specific physiological and pathological conditions. Among the most well-known autophagy activators are rapamycin, resveratrol, metformin, and trehalose with their mechanisms of action extensively reviewed elsewhere [26]. A recent study demonstrated that systemic administration of rapamycin, a well-known inhibitor of mTOR, enhanced muscle regeneration in mice with skeletal muscle dysfunction. This effect was attributed to the ability of rapamycin to activate autophagy in muscle stem cells, thereby improving their activation, proliferation, and myogenic potential [207]. Similarly, rapamycin-induced autophagy promoted peripheral nerve regeneration after sciatic nerve crush injury in rats [208]. In line with these findings, resveratrol, a natural compound found in the skin of red grapes and a potent autophagy inducer via mTOR inhibition and AMPK activation [209,210], has demonstrated regenerative benefits across multiple models. It promoted liver repair in mice following drug-induced liver damage [211], as well as axonal and peripheral nerve regeneration after injury [212,213]. Interestingly, in rodents, resveratrol enhanced periodontal bone regeneration by improving the function and regenerative capacity of mesenchymal stem cell aggregates, a promising strategy in regenerative medicine [214]. Similarly, metformin, an antidiabetic agent and a well-known autophagy inducer via several signaling pathways, including the AMPK/mTOR pathway and the activation of different sirtuins (e.g., SIRT 1) [215], has demonstrated regenerative benefits across various models. In zebrafish, metformin accelerates heart regeneration by inducing autophagy [216]. In rodent models, metformin administration promoted axonal regeneration following injury [217,218]. As expected, autophagy inhibition leads to impaired tissue regeneration across various animal models and experimental conditions. For instance, pharmacological inhibition of autophagic flux in *Hydra* using Bafilomycin A1, a well-established inhibitor of the late phase of autophagy, or with the ULK1 inhibitor SBI-0206965, effectively blocked head regeneration in these highly regenerative organisms [219]. Similarly, in planarians, treatment with the autophagy inhibitor 3-methyladenine (3-MA), a PI3K inhibitor that disrupts autophagosome formation, significantly impaired the regeneration of both head and tail structures [220]. Considering this, and given that both planarians and *Hydra* have the extraordinary ability to undergo whole-body regeneration, understanding how these organisms regulate autophagy to achieve complete regeneration represents an exciting avenue for future research. However, the study of autophagy in these organisms is still in its early stages. A deeper understanding of this process could reveal novel regulatory mechanisms and signaling pathways that may be harnessed for developing therapeutic interventions aimed at regenerating human organs and tissues, rather than merely repairing damage, with all the associated biological and clinical implications. In other models such as in zebrafish, autophagy inhibition using chloroquine (CQ), another well-characterized inhibitor of autophagic flux, counteracted the pro-regenerative effects of *Rehmanniae Radix Praeparata* (RRP), a traditional Chinese medicine, during fin regeneration following amputation [221]. In mammalian systems, the inhibition of autophagy with 3-MA impaired muscle regeneration in mice following myotoxic injury [69] and reduced nerve regeneration in rats after nerve crush injury [208]. Similarly, Bafilomycin A1 treatment disrupted axon regeneration in *C. elegans* following laser-induced axotomy [222]. Collectively, these studies highlight the potential of autophagy modulation as a promising therapeutic strategy for enhancing the regeneration of specific tissues and organs (Figure 5). Targeting the autophagy pathway holds significant promise for preventing or reversing human diseases associated with the limited regenerative capacity of our tissues.

### Autophagy as a Target for Mammalian Skin Wound Healing

Skin wound healing is a crucial repair process in mammals, and autophagy has been extensively studied for its significant role in regulating this complex biological response [223,224]. For example, autophagy has been shown to promote the migration and differentiation of keratinocytes during the pro-healing effects of fibroblast growth factor 21 (FGF21) in skin wounds [140]. Notably, FGF21 induces autophagic flux through TFEB activation, contributing to regulating extracellular matrix degradation [225], and has also been implicated in facilitating nerve regeneration following sciatic nerve injury [226]. Similarly, autophagy has been found to enhance vascularization and wound healing, through the paracrine secretion of vascular endothelial growth factor (VEGF) in a full-thickness cutaneous wound mice model [227]. Furthermore, autophagy has been reported to elevate the levels of multiple growth factors, including basic fibroblast growth factor (bFGF), epidermal growth factor (EGF), connective tissue growth factor (CTGF), transforming growth factor-beta (TGF-β), and VEGF in skin tissues. This upregulation may underlie the pro-healing effects of autophagy during tissue expansion, a technique in which mechanical stretch stimuli promote skin wound healing [113]. Another study revealed that autophagy in keratinocytes is necessary not only for keratinocyte migration but also for their proliferation, as well as for the activation of dermal fibroblasts and the recruitment of immune cells, including macrophages, neutrophils, and mast cells, which collectively facilitate wound healing in mice [82]. Consequently, several pharmacological strategies targeting autophagy activation have been explored to accelerate wound healing. For instance, rapamycin treatment in a rat model of burn wounds facilitated wound healing by increasing autophagy activity [228]. Similarly, resveratrol accelerated skin wound healing in murine models subjected to various types of injuries and cutaneous wounds [229,230,231,232]. Topical application of metformin has also been shown to significantly enhance skin repair and wound healing across different cutaneous wound models [229,233,234]. Trehalose, a natural disaccharide, and potent autophagy enhancer [235] via TFEB activation [236] or blocking glucose transport [237,238,239] demonstrated pro-regenerative effects in cutaneous wound healing. These effects are mediated through the induction of a pro-healing senescence-like state in fibroblasts [240] and the upregulation of key autophagy proteins, including ATG5 and ATG7 [241]. Epigallocatechin gallate (EGCG), the primary active compound in green tea, facilitated wound healing in diabetic rats by stimulating autophagy in keratinocytes, which in turn promoted their migration and proliferation [242]. Another study in mice demonstrated that exosomes derived from stem cells from human exfoliated deciduous teeth (SHED) enhanced wound healing while reducing itching, primarily by stimulating macrophage autophagy [243]. However, although the previous studies highlight the beneficial role of autophagy in wound healing, other findings suggest contrasting effects. For instance, the deletion of the autophagy-related gene *Atg7* in mice endothelial cells has been shown to enhance skin wound healing, accompanied by increased recruitment of macrophages and lymphocytes during the early phases of the healing process [244]. Similarly, inhibition of autophagy with 3-MA administration significantly improved wound healing in a mouse model of full-thickness wounds by modulating the YAP/IL-33 signaling pathway [97]. Additionally, inhibition of autophagic flux through bafilomycin treatment in diabetic mice accelerated wound healing by promoting cell proliferation, enhancing collagen production, and regulating the inflammatory response [245]. In another study, autophagy induced by advanced glycation end products (AGEs) was found to promote macrophage polarization to the pro-inflammatory M1 phenotype, leading to excessive inflammation and impaired wound healing in diabetic mouse skin and patients with chronic wounds. Notably, inhibition of autophagy with 3-MA completely rescued the delayed wound healing caused by AGEs [246]. Furthermore, other studies have reported that rapamycin treatment impairs wound healing in both mice [247] and patients with cutaneous carcinogenesis [248]. In summary, this evidence highlights the complex and context-dependent role of autophagy in wound healing. This complexity requires a precise understanding of which phases of the wound healing process, or which specific cell types, benefit from either the activation or inhibition of autophagy at a given time. A promising avenue for exploration may involve the development of strategies that combine autophagy inducers and inhibitors, similar to those already employed in cancer therapy [249]. Such innovative approaches could pave the way for more effective and targeted therapeutic strategies for wound healing in humans, ultimately making a significant impact on healthcare and society.

## 12. Conclusions

Together, there is data strongly suggesting that the autophagy pathway is critical in promoting various cellular mechanisms comprising tissue regeneration and repair across regenerative species (summarized in Figure 4). Autophagy contributes to the regeneration process by: (1) helping in the sensing and response to injuries and tissue damage; (2) facilitating an immediate immune response and promoting the clearance of invading pathogens following tissue injury; and (3) supporting the cell cycle re-entry, proliferation and migration of cells in response to damage. Furthermore, we anticipated the connection between autophagy and cellular senescence, emphasizing the importance of investigating the contribution of the autophagy machinery in inducing a pro-healing senescent state during tissue regeneration. Pharmacological modulators of autophagy have demonstrated effectiveness in promoting and accelerating the repair and regeneration of damaged tissues. These findings suggest that targeting autophagy could serve as a viable strategy to modulate tissue repair and regeneration, underscoring its potential relevance for therapeutic applications in regenerative medicine. Although it is evident that autophagy plays a vital role in tissue regeneration, many aspects of this relationship remain poorly understood. Elucidating the molecular mechanisms by which autophagy regulates and orchestrates the diverse processes involved in regeneration holds significant potential for the development of novel therapeutic strategies aimed at preventing and even reversing human diseases, including cancer and age-related conditions.

## Figures and Tables

**Figure 1 cells-14-00282-f001:**
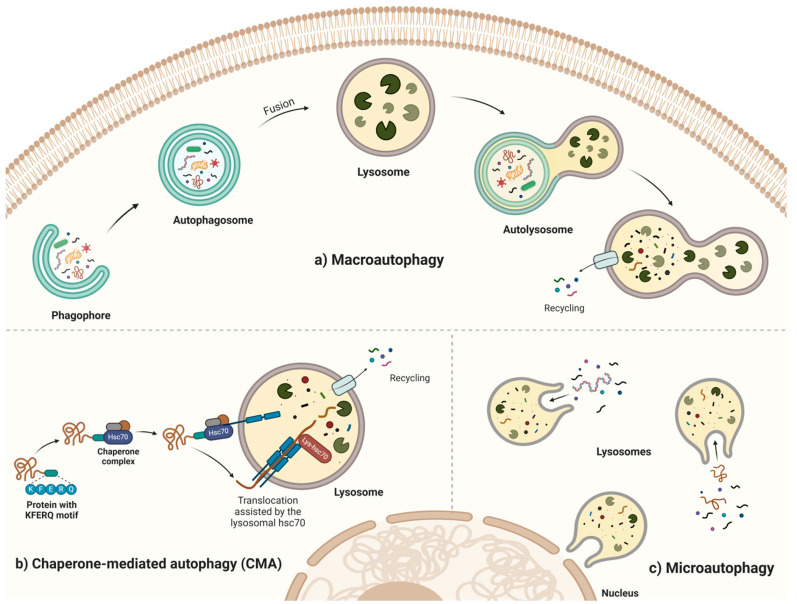
Overview of Autophagy Pathways. Autophagy utilizes distinct pathways to degrade intracellular components, including damaged proteins, organelles, and pathogens, ensuring cellular homeostasis. (**a**) Macroautophagy involves the sequestration of cytoplasmic material within a double-membrane vesicle known as the *autophagosome*. The autophagosome subsequently fuses with the lysosome, where lysosomal enzymes degrade its contents. (**b**) Chaperone-mediated autophagy (CMA) selectively targets individual proteins containing the KFERQ-like motif. These proteins are recognized by the chaperone Hsc70, transported to the lysosomal membrane, and translocated into the lysosomal lumen through the receptor LAMP2A for degradation. (**c**) Microautophagy involves the direct uptake of cargo by lysosomes or late endosomes through membrane invagination or protrusion. Once internalized, the autophagic cargo is delivered to the lysosomal lumen, where it undergoes enzymatic degradation. Created in BioRender. Gonzalez Estevez, C. (2025) https://BioRender.com/a77a953.

**Figure 2 cells-14-00282-f002:**
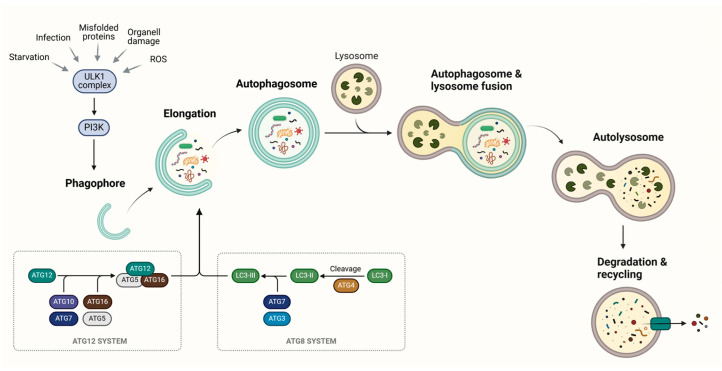
Key Molecular Steps in the Autophagy Pathway. Autophagy is initiated in response to various stimuli, such as nutrient deprivation, misfolded proteins, reactive oxygen species (ROS), or infections. These signals activate the ULK1 complex, which triggers the formation of the isolation membrane (phagophore) in the cytosol. As the phagophore elongates, cytosolic cargo—including misfolded proteins, damaged organelles, and pathogens—is sequestered through the coordinated action of ATG proteins, which function as a ubiquitin-like conjugation system. Upon elongation and closure, the phagophore matures into a double-membrane vesicle known as the autophagosome. The autophagosome then fuses with the lysosome to form the autolysosome, where lysosomal hydrolytic enzymes degrade the cargo and the inner autophagosomal membrane, facilitating recycling of the breakdown products. Created in BioRender. Gonzalez Estevez, C. (2025) https://BioRender.com/d35c194.

**Figure 3 cells-14-00282-f003:**
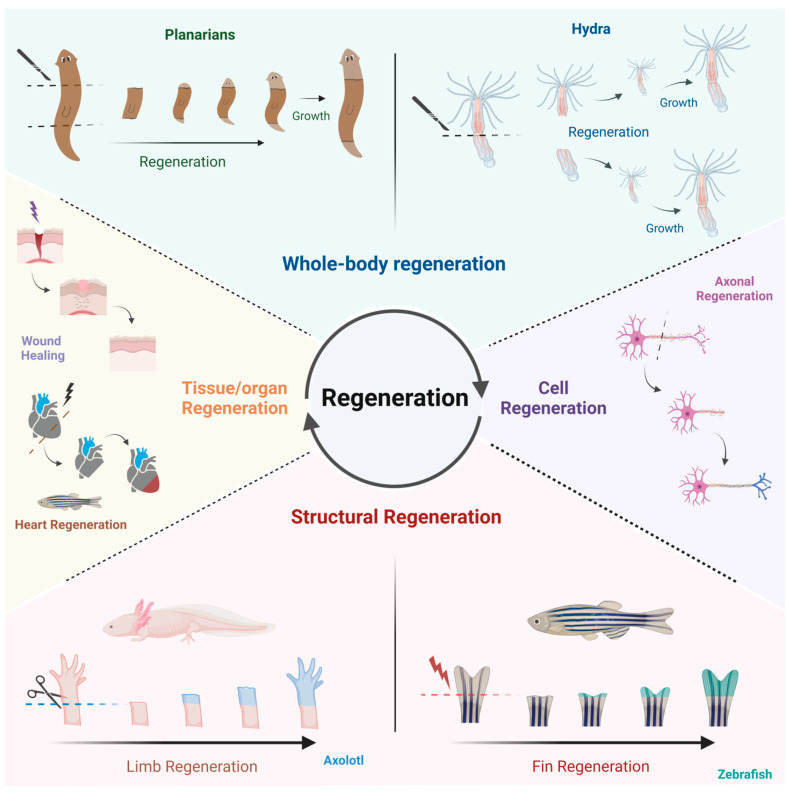
Regeneration. Regeneration is a biological process in which organisms restore lost tissues, structures, or even entire bodies. Certain species, such as planarians and Hydra, exhibit whole-body regeneration, where they can regenerate an entire organism from a small body fragment. *Structural regeneration occurs* after limb or fin resection in axolotl and zebrafish, respectively. Some organs, such as our skin after superficial damage (deep wounds can only be repaired) and zebrafish heart after cryoinjury or transection, possess notable regenerative capacities, allowing them to restore their structure and function following injury. Additionally, specific cells, such as neurons, can regenerate damaged components, like axons, through a process known as *axonal regeneration*. Created in BioRender. Gonzalez Estevez, C. (2025) https://BioRender.com/n86y859.

**Figure 4 cells-14-00282-f004:**
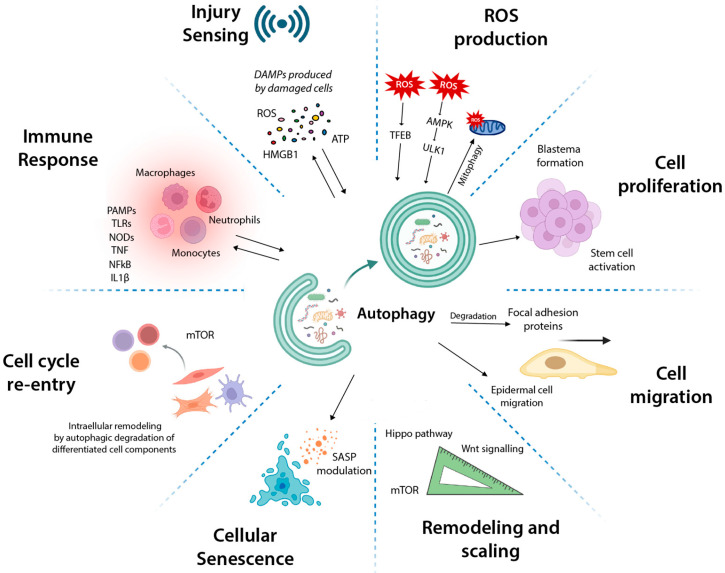
Autophagy: a Potential Player in Regeneration. In this model, autophagy serves as a central mechanism driving tissue regeneration and repair across diverse regenerative species. It orchestrates multiple processes critical to regeneration, including injury sensing and damage response, rapid immune activation, and clearance of invading pathogens. Autophagy further promotes cellular proliferation, migration, and cell cycle re-entry necessary for tissue repair. Additionally, it facilitates tissue remodeling to ensure correct proportions and size during regeneration. Importantly, autophagy also regulates cellular senescence, a key biological process in successful tissue regeneration. Created in BioRender. Gonzalez Estevez, C. (2025) https://BioRender.com/n98d270 and edited with Adobe Illustrator.

**Figure 5 cells-14-00282-f005:**
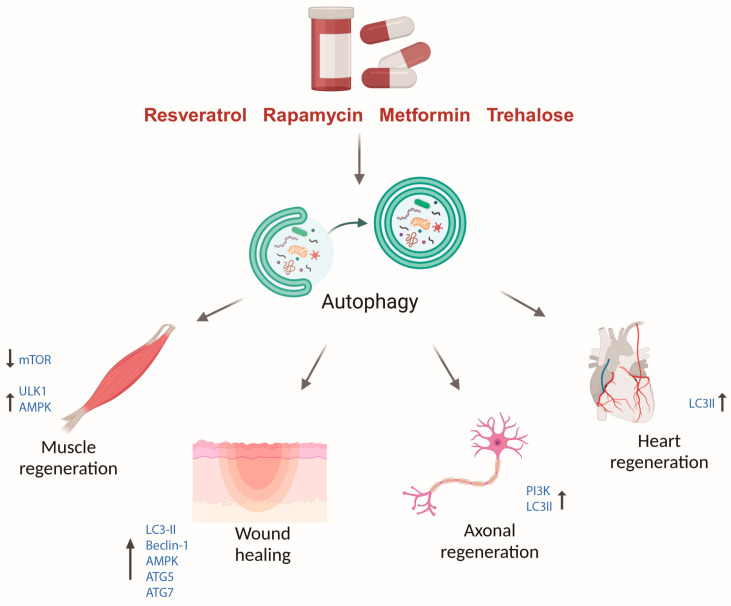
Pharmacological Modulation of Autophagy Enhances Tissue Regeneration. Pharmacological targeting of the autophagy pathway represents a promising therapeutic strategy for promoting tissue repair and regeneration due to the critical role of autophagy in regulating various components of the regenerative process. Several well-established autophagy inducers, including rapamycin, resveratrol, metformin, and trehalose, have demonstrated significant efficacy in enhancing muscle regeneration, accelerating wound healing, and promoting axonal and cardiac tissue repair. Created in BioRender. Gonzalez Estevez, C. (2025) https://BioRender.com/y97n890 and edited with Adobe Illustrator.

**Table 1 cells-14-00282-t001:** The table outlines the primary biological processes involved in regeneration and repair, along with their associated consequences and the key molecules that play a role. It is important to note that not all processes occur in every tissue and/or organism, and the sequence of events can vary depending on the specific regenerative context. Additionally, there may be some overlap between processes. The referenced sources provide excellent reviews covering a variety of regeneration models.

Biological Process	Consequences of the Biological Processes	Key Molecules and Mediators
Sensing the injury [11,12,13].	Loss of tissue integrity, infection, DNA damage, cell death, senescence.	DAMPs (ROS, ATP, PUFAs, Egr), Ca^2+^, Wnt pathway, p38 signaling, MAPK/ERK pathway, regeneration-specific genetic programs through TREEs and re-activation of embryonic genetic programs, metabolic re-wiring.
Immune response [14].	Macrophage activation, cell migration, inflammation, cell death, debris clearance, defense from pathogens, induction of proliferation/differentiation, fibrosis, and scar formation.	PAMPs, Toll-like receptor signaling, NOD-like receptor signaling, signaling from macrophages.
Peripheral barrier restoration [15].	Re-epithelialization and extracellular matrix (ECM) remodeling for wound closure.	Cytokines and growth factors, Ca^2+^, matrix metalloproteinase (MMPs), integrins.
Cell activation (may involve migration), cell cycle re-entry, and proliferation [16].	Formation of a blastema or wound repair.	Wnt, BMP, Hippo, Junk, and IGF pathways, mTOR, Akt, SMG-1, growth factors, signaling from apoptotic and senescent cells.
Differentiation and morphogenesis [17,18,19].	Cell fate specification, pattern, and shape formation, and integration with existing tissue.	Wnt and BMP pathways, metabolic re-wiring.
Remodeling and scaling [20,21,22,23].	Growth with or without morphogenesis.	mTOR, JNK, and Hippo pathways, Wnt signaling, STRIPAK.

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
