# Peer review of "Autophagy in Tissue Repair and Regeneration"

_cells, 2025, doi:10.3390/cells14040282_

Round 1
Reviewer 1 Report
Comments and Suggestions for Authors
The paper focuses on the essential importance of autophagy in tissue regeneration and repair, which is an important topic and timely. But some of the most important points should be discussed in details in order to make the review more enlightened and comprehensive. In particular, the paper would benefit from exploring autophagy as a target for wound healing, including the effect of autophagy modulators like rapamycin or chloroquine on keratinocyte movement, fibroblast function and angiogenesis in chronic wounds. It is also important to talk about the connection between autophagy and cellular senescence in tissue repair, and especially how autophagy regulates senescence-associated secretory phenotypes (SASP) and how this affects repair and fibrosis. This paper has not paid enough attention to lipid metabolism and autophagy (lipophagy) in the energy-demanding repair of tissues, such as the liver and muscle, which are fundamental to understanding energy balance during regeneration. Additionally, organ-specific information about autophagy in regeneration, including its unique mechanisms for cardiac, liver, or neural tissue repair, needs to be brought into focus to include more people (this is mentioned in the manuscript submission but remains superfacial. I encourage the esteemed authors to focus on details, instead of just introducing in some short way). Adding a section focusing on autophagy-growth factor crosstalk (trellox, VEGF, FGF) and its impact on cellular and extracellular matrix remodeling in repair would further solidify the manuscript (one new section). Crucially, the review should include a separate chapter on secretory autophagy and inflammation, explaining how autophagy-mediated secretion impacts inflammatory signals that are essential for tissue repair. If you want to make it more accessible, include schematics or diagrams for each section indicating key pathways and mechanisms. Lastly, the inclusion of a section on clinical trials investigating autophagy modulators for tissue repair will make the manuscript more translational, moving the translation from preclinical to clinical research. These changes will significantly expand the breadth and impact of the manuscript.
I appreciate if the esteemed authors pay attention of every comment and respond in detail.
Reviewer 2 Report
Comments and Suggestions for Authors
In the manuscript by Moreno-Blas et al., authors summarize the literature address to interplay autophagy and tissue repair and regeneration. They highlight that autophagy could be the basis for some biological process implicated in the tissue repair and emphasizes the autophagy as a potential therapeutic target. This manuscript is well written and the topic is interesting. Here my comments:
1. Usually, at the end of the introduction should be clearly stated what is the aim and the importance of this report beyond what other papers on the same topic have previously identified.
2. In point 3 authors have focused in define regeneration process; in my opinion authors also must give a clear definition of tissue repair.
3. In points 4.1 and 5 authors fail to associate these molecular processes (DAMPs and Immune response) with tissue repair and regeneration.
4. In point 10, authors also include information about autophagy activators. What is the effect of autophagy inhibitors in these biological processes?
5. Is it possible that authors modify the Figure 5 to include the target molecular mechanisms of autophagy modulation?
Reviewer 3 Report
Comments and Suggestions for Authors
Although many comprehensive reviews regarding autophagy and medicine (particularly for cancer) have been done in the literatures, in this review, PIs try to further overview and focus on evaluating the evidence supporting the involvement of the autophagy pathway in the regeneration process. They explore how modulating autophagy can enhance or accelerate tissue regeneration and wound healing, highlighting autophagy as a promising therapeutic target in regenerative medicine. The review also suggests valuable insights into potential therapeutic applications for regenerative medicine, aiming to harness or manipulate autophagy to improve tissue repair and regeneration in humans. Some points are noted below:
1. When we say “Dysregulation of autophagy”, means that either excessive or deficiency. In most part appears excessive autophagy (more)? How is less autophagy deficiency in tissue repair and regeneration, please discuss this point?
2. As mention above, it is known that “Combination of an Autophagy Inducer and an Autophagy Inhibitor: A Smarter Strategy Emerging in Cancer Therapy”. How Rapamycin; 3MA; CQ; Baf A1 or some natural compounds that affect autophagic flux to cause tissue repair and regeneration. Also discuss this point.
3. For the development of autophagy-based treatments of tissue repair and regeneration. In the perspective: how can we learn from planarian and Hydra for whole body regeneration? Need deeply compare to induced pluripotent stem cells and wound tissue level.
4. In pages 250-280 Toll-like receptors (TLRs) and other pattern recognition receptors (PRRs) recognize these danger signals and initiate a complex inflammatory cascade. Toll-like family (TLR1, TLR2, TLR4, TLR5, TLR6, and TLR10 are located on the cell membrane, whereas TLR3, TLR7, TLR8, and TLR9 are located in intracellular vesicles). Only TLR2, 3, 4 and 6 were mentioned here, what others in tissue repair and regeneration?
Reviewer 4 Report
Comments and Suggestions for Authors
Dear Authors
The review titled "Autophagy in tissue repair and regeneration” provides a comprehensive exploration of autophagy's role in tissue regeneration across various species. It highlights the potential of modulating autophagy to enhance or accelerate regeneration and wound healing, underscoring its promise as a therapeutic target in regenerative medicine. While the topic is highly engaging and relevant, several areas require improvement to enhance its clarity, depth, and utility.
Here are the comments for improvement:
- The review effectively addresses mechanisms related to tissue repair and regeneration under conditions such as infection, injury, and cancer. However, it tends to conflate these different contexts, which diminishes the clarity and specificity of the analysis. To address this, the review should be reorganized to distinctly categorize and analyze mechanisms based on the type of injury or pathological situation.
- Certain sections, particularly Section 4, would benefit from a more detailed examination of the mechanisms described. This should include a more comprehensive discussion of cellular processes like autophagy's interplay with inflammation, as inflammation is a key component of tissue repair and regeneration that is currently underexplored in this section.
- Figures, especially Figure 4, should be enhanced to include more detailed information about the mechanisms involved in the processes discussed. Incorporating visual annotations or flowcharts that connect specific molecules, pathways, and outcomes in repair and regeneration could significantly improve clarity and value.
To aid readers, the authors should include tables summarizing key mechanisms associated with tissue repair and regeneration. These tables should detail: the principal pathways or processes involved; key molecules and mediators, the consequences of these mechanisms; and the relevant references supporting each finding. Sections 4, 5, and 6, in particular, would be strengthened by the inclusion of such tables, providing a concise yet informative overview of the key points discussed.
By addressing these points, the review could provide a more structured and nuanced analysis of autophagy's involvement in tissue repair and regeneration, making it more robust and valuable.
Round 2
Reviewer 1 Report
Comments and Suggestions for Authors
The authors have sufficiently answered my comments. I do not have any further comments